

# The land flatworm *Amaga expatria* (Geoplanidae) in Guadeloupe and Martinique: new reports and molecular characterization including complete mitogenome

Jean-Lou Justine[1], Delphine Gey[2], Jessica Thévenot[3],
Romain Gastineau[4] and Hugh D. Jones[5]

[1] ISYEB - Institut de Systématique, Évolution, Biodiversité, Muséum National d'Histoire Naturelle, Paris, France
[2] Molécules de Communication et Adaptation des Micro-Organismes, Muséum national d'Histoire naturelle, Paris, France
[3] Patrinat, Muséum national d'Histoire naturelle, Paris, France
[4] Institute of Marine and Environmental Sciences, University of Szczecin, Szczecin, Poland
[5] Life Sciences Department, Natural History Museum, London, UK

Corresponding author
Jean-Lou Justine, justine@mnhn.fr

## ABSTRACT

**Background:** The land flatworm *Amaga expatria* Jones & Sterrer, 2005 (Geoplanidae) was described from two specimens collected in Bermuda in 1963 and 1988 and not recorded since.

**Methods:** On the basis of a citizen science project, we received observations in the field, photographs and specimens from non-professionals and local scientists in Martinique and Guadeloupe. We barcoded (COI) specimens from both islands and studied the histology of the reproductive organs of one specimen. Based on Next Generation Sequencing, we obtained the complete mitogenome of *A. expatria* and some information on its prey from contaminating DNA.

**Results:** We add records from 2006 to 2019 in two French islands of the Caribbean arc, Guadeloupe (six records) and Martinique (14 records), based on photographs obtained from citizen science and specimens examined. A specimen from Martinique was studied for histology of the copulatory organs and barcoded for the COI gene; its anatomy was similar to the holotype, therefore confirming species identification. The COI gene was identical for several specimens from Martinique and Guadeloupe and differed from the closest species by more than 10%; molecular characterisation of the species is thus possible by standard molecular barcoding techniques. The mitogenome is 14,962 bp in length and contains 12 protein coding genes, two rRNA genes and 22 tRNA genes; for two protein genes it was not possible to determine the start codon. The mitogenome was compared with the few available mitogenomes from geoplanids and the most similar was *Obama nungara*, a species from South America. An analysis of contaminating DNA in the digestive system suggests that *A. expatria* preys on terrestrial molluscs, and citizen science observations in the field suggest that prey include molluscs and

earthworms; the species thus could be a threat to biodiversity of soil animals in the Caribbean.

# INTRODUCTION

The land flatworm *Amaga expatria Jones & Sterrer, 2005* (Platyhelminthes, Geoplanidae) was described from two specimens collected in Bermuda in 1963 and 1988 (*Jones & Sterrer, 2005*). It has not been recorded since. *Jones & Sterrer (2005)* concluded that the species was an alien species in Bermuda, probably introduced from South America since other members of the genus have been collected in this region, including Colombia, Peru, Chile, Brazil, Paraguay and Argentina (*Ogren & Kawakatsu, 1990*); the genus *Amaga Ogren & Kawakatsu, 1990* currently includes 10 species (*Grau et al., 2012*).

In 2003, one of us (JLJ) undertook a citizen science program in France about alien land flatworms. Records were unexpectedly received from other locations including French overseas territories (*Justine et al., 2014*, *2015*, *2018a*, *2018b*, *2019*; *Justine & Winsor, 2020*). Among these were several records of large land flatworms from the two Caribbean islands of Martinique and Guadeloupe with similar dimensions and pigmentation to *A. expatria* and tentatively identified as such. To confirm this identification, one specimen from Martinique has been partially sectioned for anatomical comparison with the type material. The same specimen was also subjected to molecular sequencing (COI barcoding). Another specimen from Martinique was included in a comparative study of the mitogenome of several land flatworms (*Gastineau & Justine, 2020*; *Gastineau et al., 2019*, *2020*).

We present here new records of *A. expatria* in two islands of the Caribbean, and provide additional morphological information, the first barcoding characterisation of the species and its complete mitogenome.

# MATERIALS AND METHODS

## Citizen science and collection of information

Records were collected from 2013 to 2019, a period of 7 years (single records from 2006 to 2012 are also included). We used the same methods as for our previous research on land flatworms (*Justine et al., 2014*, *2015*, *2018a*, *2018b*, *2019*). A blog (*Justine, 2019*) and a twitter account (@Plathelminthe4) were the main tools for collecting and transmitting information. The collaboration of local natural history associations and of the FREDON (Regional federations for the control of pests) in the departments of Martinique and Guadeloupe was also instrumental. Reports of sightings were received from the general public and from professionals, generally by email. We solicited and obtained specimens. Specimens were obtained alive, fixed in near boiling water and preserved in 95% ethanol, or sometimes fixed directly in cold ethanol. They were sent to the Muséum

National d'Histoire Naturelle (MNHN) in Paris, registered and processed for molecular studies.

## Histology

A specimen from Martinique, MNHN JL146, was used for histology. It was killed alive in boiling water, then kept in 80% ethanol. A portion about 1.7 cm long containing the copulatory apparatus was removed for sectioning. Horizontal longitudinal sections (HLS) were cut at 12 μm thickness, mounted on 41 slides, stained in haematoxylin and eosin and mounted in Canada balsam (slides 1–5 and 38–41 remain unstained in wax). Slides are deposited in the MNHN, Paris, registration number MNHN JL146.

## Molecular barcoding

For molecular analysis, a small piece of the body (1–3 mm$^3$) was taken from the lateral edge of ethanol-fixed individuals. Extraction of DNA and PCR were performed as in previous similar works (*Justine et al., 2019*). Briefly, a fragment of 424 bp was amplified with the primers JB3 (=COI-ASmit1) and JB4.5 (=COI-ASmit2) (*Bowles, Blair & McManus, 1995*; *Littlewood, Rohde & Clough, 1997*), and a fragment of 825 bp was amplified with the primers BarS (*Álvarez-Presas et al., 2011*) and COIR (*Lázaro et al., 2009*; *Mateos et al., 2013*). PCR products were purified and sequenced in both directions on a 96-capillary 3730xl DNA Analyzer sequencer (Applied Biosystems, Foster City, CA, USA). Results of both analyses were concatenated to obtain a COI sequence of 903 bp in length. Sequences were edited using CodonCode Aligner software (CodonCode Corporation, Dedham, MA, USA), compared to the GenBank database content using BLAST, and deposited in GenBank under accession number MT602619–MT602626.

## Next generation sequencing, phylogeny and identification of contaminant DNA

A slice of flesh from specimen MNHN JL305 was sent to the Beijing Genomics Institute (BGI) in Shenzhen, which provided DNA extraction and sequencing. Sequencing was performed on a DNBSEQ platform. A total of ca. 60 million clean paired-end reads were obtained. Reads were assembled using SPAdes 3.14.0 (*Bankevich et al., 2012*) with a k-mer of 85. Contigs corresponding to the mitogenome and the nuclear ribosomal RNA genes were retrieved by customized blastn command line analyses, using already available sequences downloaded from GenBank as a custom database; ribosomal RNA genes (not used in this article), were deposited into GenBank as MT860713 (18S) and MT860719 (partial 28S). In addition to the sequences related to *A. expatria*, other positive matches belonging to contaminant DNA were obtained as explained in the results. tRNA were identified using tRNA-scan (*Lowe & Chan, 2016*). The mitogenome was verified using the Consed package (*Gordon, Abajian & Green, 1998*), and gene identification was performed using MITOS (*Bernt et al., 2013*). The genomic map was drawn using OGDRAW (*Lohse et al., 2013*). Amino acid sequences of the protein coding genes were concatenated following a protocol already described (*Gastineau & Justine, 2020*;
*Gastineau et al., 2019, 2020*), and aligned with corresponding sequences from other species using MAFFT (*Katoh & Standley, 2013*); we used the command-line version of MAFFT, with the option "G-INS-I". A maximum likelihood phylogeny was inferred from this alignment using RaxML version 8.0 (*Stamatakis, 2014*) using the MtArt substitution model (*Abascal, Posada & Zardoya, 2007*). The best tree out of 100 was computed for 100 bootstrap replicates.

### COI trees and distances

MEGA7 (*Kumar, Stecher & Tamura, 2016*) was used to evaluate distances, and construct trees. For the outgroup, we chose a sequence in GenBank from the South American species *O. nungara Carbayo et al., 2016* (MN529572) which had a 100% query cover with our sequences.

## RESULTS

### Information obtained from citizen science and other scientists

We obtained 14 verified records from Martinique (map in Fig. 1), from 2006 to 2018, and six verified records from Guadeloupe, from 2012 to 2018 (map in Fig. 2). Records were generally obtained as photographs, but we also received five specimens from Martinique and three specimens from Guadeloupe (Table 1). In addition, François Meurgey (email, 29 January 2016) added information about Guadeloupe: "*Amaga expatria* is quite common in the high series of the mesophilic forest and between 400 and 700 m altitude in the moist forest. I met it in the municipalities of Baillif (St. Louis river), Matouba, Trois-Rivières, and Gourbeyre. It seems more frequent in the South of Basse-Terre. I have observed it attacking snails of the genera *Helicina*, *Pleurodonte* and earthworms." Laurent Charles (email, 15 May 2020) sent a photograph showing predation on a snail identified as *Helicina platychila* (Von Mühlfeld, 1824).

### Morphology and histology

Live specimens mentioned in this study, measured on photographs (Figs. 3–5), were 128–132 mm in length and 5.5–9 mm in width in extended state, and 35 × 12 mm in contracted state.

Description of Specimen MNHN JL146.

Living dimensions (Fig. 3): length 128 mm, width 5.5 mm. Preserved dimensions: length 108 mm, width 9 mm, height 2 mm; mouth 53%; genital pore 75%.

The copulatory apparatus (Fig. 6) is about 8 mm long from the anterior of the male system to the posterior of the female system. The male system is about 5 mm long and the female system about 3 mm long.

Two sperm ducts, about 1.2 mm apart, each with copious stored sperm (cyanophylic) approach the copulatory apparatus, briefly turn anteriorly before opening separately into the ventral end of a single duct (Fig. 6D). This duct has a thick muscular wall and runs vertically from ventral to dorsal for about 1080 μm (runs through 90 sections × 12 μm). At its dorsal end this duct continues posteriorly as a narrow sinuous duct before broadening into the ejaculatory duct. The ejaculatory duct terminates in a short penis

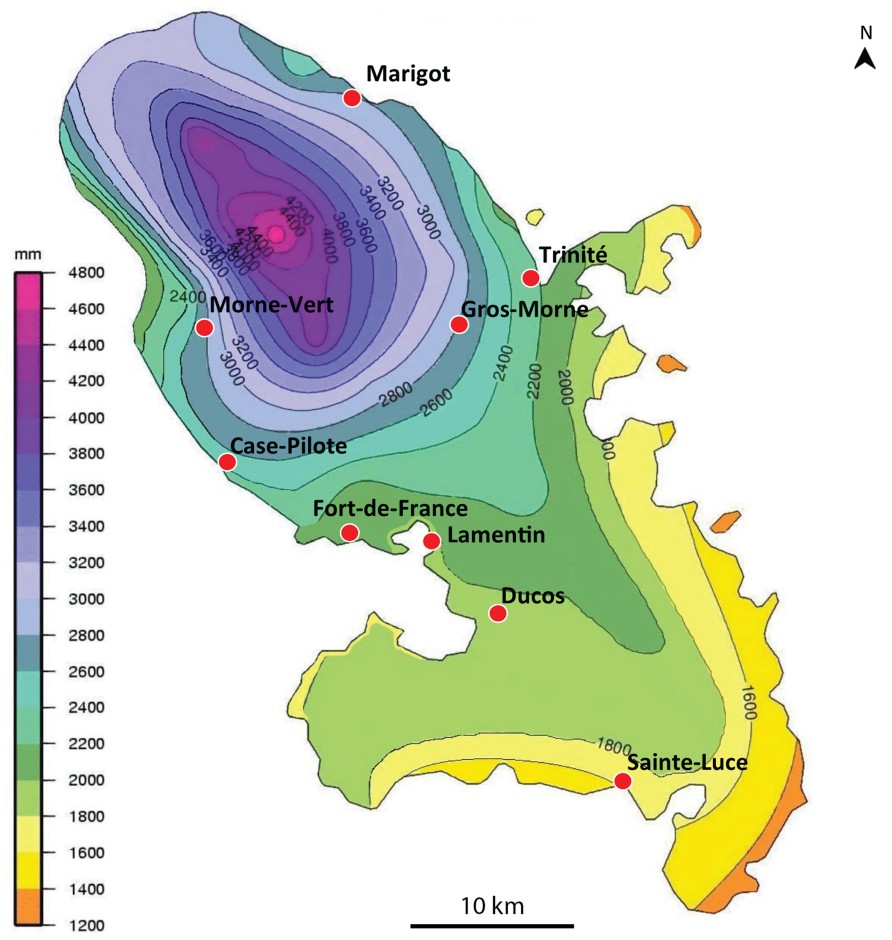

**Figure 1 *Amaga expatria*, map of records in Martinique.** The background colours indicate annual raindrop falls. Most records are from the Northern part of the island where raindrops are high, but the record in Sainte-Luce in the South is a from a relatively drier part. Map by Jessica Thévenot, background provided by Météo-France and used with authorisation.

about 800 µm long and 600 µm wide (Figs. 6A and 6B). There are atrial folds outside the penis in the common antrum.

The two ovovitelline ducts (Figs. 6A and 6B) are about 2 mm apart anterior to the penis, they run posteriorly and at about the level of the penis they turn dorsally to join and open into the combined female duct. Copious shell glands (eosinophilic) are present and open into both ovovitelline ducts before they join (Fig. 6C) to form the combined female duct. The combined female duct broadens, and has one or two longitudinal folds, before opening into the common antrum.

The gonopore opens from the common antrum via a short, narrow duct.

## Molecular characterization—COI

For four specimens, the amplified COI sequences obtained were identical along their whole length (903 bp). These specimens were JL289, JL305 (obtained both from Sanger sequencing and from the mitogenome) and JL310 from Martinique, and JL319 from

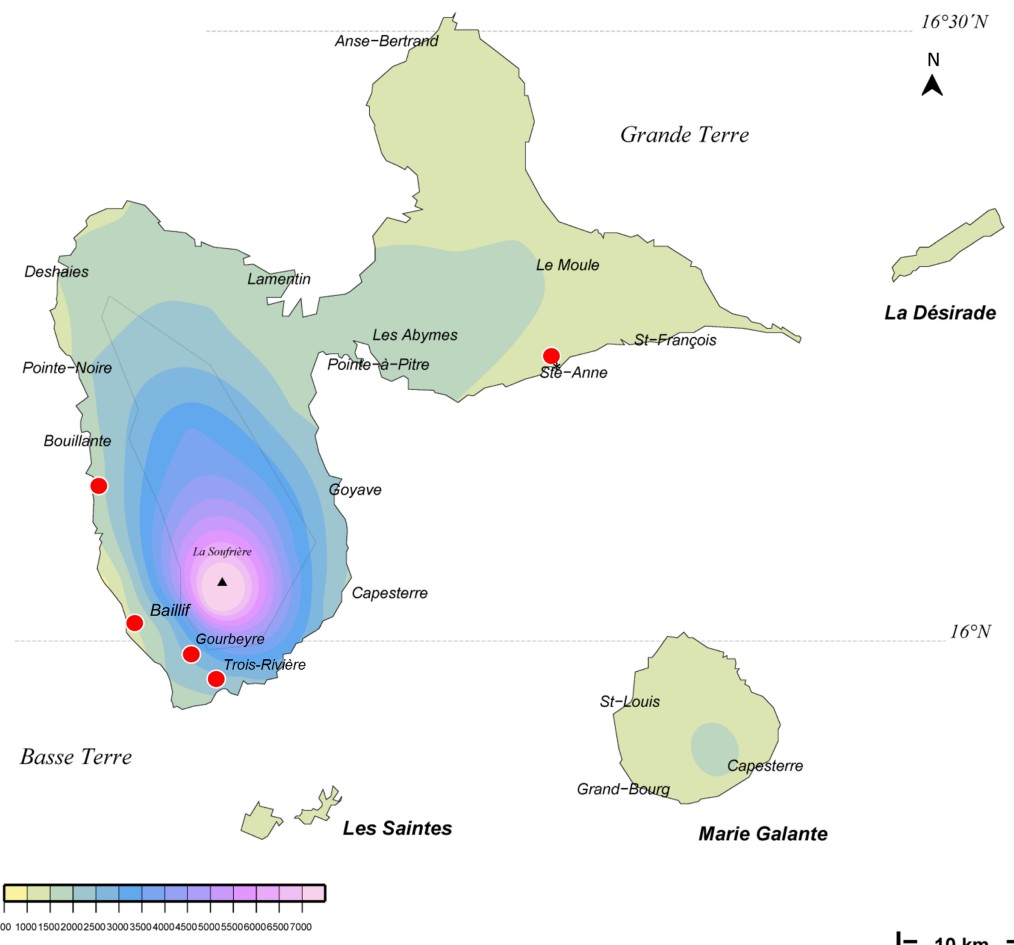

**Figure 2 _Amaga expatria_, map of records in Guadeloupe.** The background colours indicate annual raindrop falls. Most records are from Basse Terre, where raindrops are high, but the record in Ste-Anne in Grande Terre is from a relatively drier part. Map by Jessica Thévenot, background provided by Météo-France and used with authorisation.

Guadeloupe. This demonstrates that the same species was found in both islands. Specimen JL146 from Martinique, which was processed for histology, and three other specimens, JL216 and JL217 from Guadeloupe and JL262 from Martinique, provided shorter sequences but these were also identical between them and with the 4 sequences above for their portion in common. This demonstrates that the specimen studied for histology is from the same species, therefore confirming that the species in both islands is _A. expatria_. Differences with _O. nungara_, calculated on the 903 bp in common, were 12%.

## Mitogenome

The mitogenome (Fig. 7) is 14,962 bp long (GenBank accession number: MT527191). It contains 12 protein coding genes, two ribosomal RNA genes and 22 transfer RNA genes. The mitogenome is completely colinear with that of _O. nungara_ (KP208777) (_Solà et al., 2015_) and similar in size (14,909 bp for _O. nungara_). A megablast query using the whole sequence of the mitogenome shows a global 83.77% identity between these

**Table 1 Records of *Amaga expatria* from Guadeloupe and Martinique.**

| Date | Record (specimen number and/or photo) | Number of specimens | Commune | Department | COI sequences | Collector/observer |
|---|---|---|---|---|---|---|
| 28 February 2006 | Photo | 0 | Case-Pilote | Martinique | No | Régis Delannoye |
| 17 August 2013 | Photo | 0 | Case-Pilote | Martinique | No | Régis Delannoye |
| 25 November 2013 | Movie | 0 | Fort de France | Martinique | No | Anonymous |
| 20 December 2013 | Photo | 0 | Le Lamentin | Martinique | No | Pierre Damien Lucas |
| 6 May 2014 | MNHN JL146 + photo | 1 | Le Gros Morne | Martinique | MT602619 | Clément Dromer |
| 21 March 2015 | Photo | 0 | La Trinité | Martinique | No | Régis Delannoye |
| 5 August 2015 | MNHN JL262 | 1 | La Trinité | Martinique | MT602622 | Olivier Palcy |
| 17 October 2015 | Photo | 0 | Le Gros Morne | Martinique | No | Pierre Damien Lucas |
| 12 November 2015 | MNHN JL305 | 1 | Le Morne Vert | Martinique | MT602624* | Mathieu Coulis |
| 13 November 2015 | Photo | 0 | Ducos | Martinique | No | Cedric Rareg |
| 4 February 2016 | MNHN JL289 | 1 | Le Marigot | Martinique | MT602623 | Régis Delannoye |
| 18 June 2017 | MNHN JL310 + photo | 1 | Le Lamentin | Martinique | MT602625 | Mathieu Coulis |
| 22 June 2017 | Photo | 0 | Fort de France | Martinique | No | Marcel Bourgade |
| 27 January 2018 | Photo | 0 | Sainte-Luce | Martinique | No | Stéphane Bras |
| 12 February 2019 | Photo | 0 | La Trinité | Martinique | No | Régis Delannoye |
| 20 December 2012 | Photo | 0 | Trois Rivières | Guadeloupe | No | Guy van Laere |
| 6 December 2014 | MNHN JL216 + photo | 1 | Gourbeyre | Guadeloupe | MT602620 | Laurent Charles |
| 12 December 2014 | MNHN JL217 + photo | 1 | Bouillante | Guadeloupe | MT602621 | Laurent Charles |
| 21 January 2016 | Photo | 0 | Baillif | Guadeloupe | No | Pierre et Claudine Guezennec |
| 20 December 2017 | MNHN JL319 + photo | 1 | Trois Rivières | Guadeloupe | MT602626 | Guy van Laere |
| 06 August 2018 | Photo | 0 | Sainte-Anne | Guadeloupe | No | Jean-Christian Rotger |

**Note:**
* For MNHN JL305, we obtained both a COI sequence (GenBank MT602624) and the complete mitogenome (GenBank MT527191).
The Table includes only observations based on photographs and specimens. François Meurgey provided additional findings in Guadeloupe: 10.10.2019, Goyave; 16.10.2019, Vieux-Fort; 22.06.2019, Petit Canal.

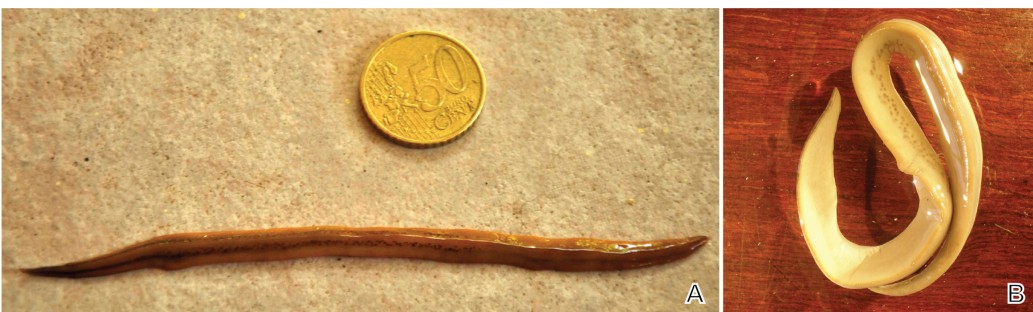

**Figure 3 *Amaga expatria*, specimen MNHN JL146 from Martinique.** (A) Living specimen, photograph by Clément Dromer; scale: the diameter of the coin is 24 mm; anterior tip is left. (B) Preserved specimen, photograph by Jean-Lou Justine. This specimen was used for anatomy.

two species. The mitogenome is also colinear with those of *Bipalium kewense*, but not with those of *Platydemus manokwari* and *Parakontikia ventrolineata*. For three genes, atp6, cox2 and ND3, it was not possible to determine the start codon. The first methionine of the predicted proteins occurs at position 72/224 for *atp6*, 112/260 for *cox2* and 44/112 for
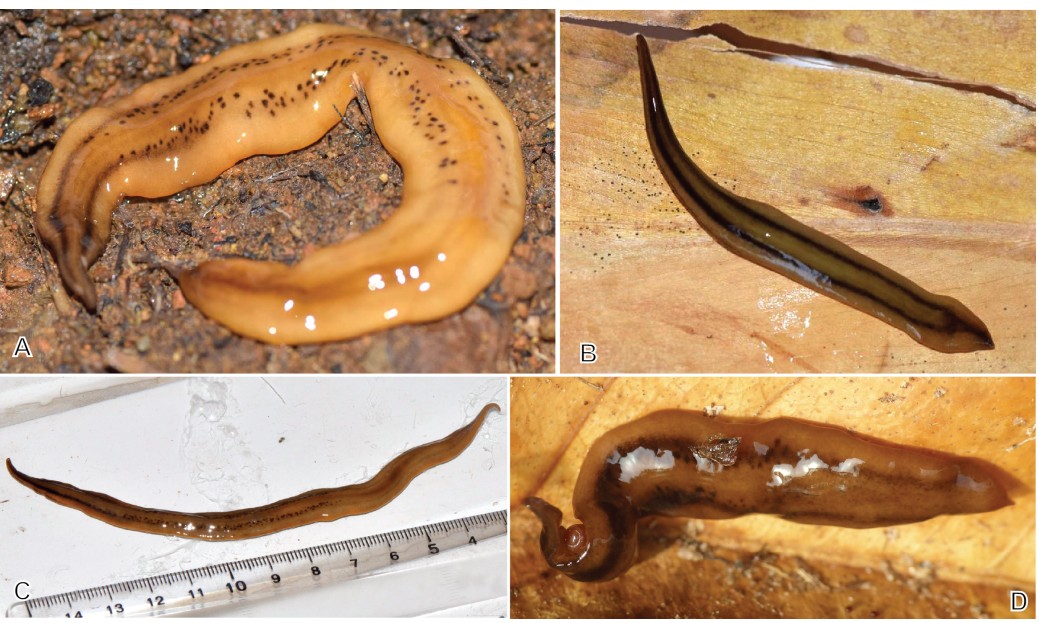

**Figure 4** ***Amaga expatria* from Martinique, living specimens.** (A) Photograph by Cedric Rareg; (B and C) Photograph by Régis Delannoye; (C) Scale in mm; (D) Photograph by Mathieu Coulis, specimen MNHN JL305. Anterior tip is left for all specimens.

*ND3*. It is worth mentioning that the impossibility to evidence a start codon for these genes was observed with *O. nungara*, but not for example with *B. kewense*, *Pl. manokwari* or *Pa. ventrolineata*. Unlike *O. nungara*, however, no overlap between the *ND4L* and *ND4* genes was evidenced.

In the tree representing a maximum likelihood phylogeny of amino acid sequences of protein coding genes, *A. expatria* is the sister-group of *O. nungara* (Fig. 8).

## Detection of an alien DNA

After assembly, sequences linked with contaminating DNA were identified. Three contigs of 3,080 bp, 7,274 bp and 15,202 bp were retrieved. Megablast analyses were performed on the NCBI portal. The best results are listed thereafter. The 3,080 bp fragment displayed a 99.66% of identity with the 18S ribosomal genes of sequences identified as the molluscs *Subulina striatella* (Rang, 1831) (MN022690), *Lissachatina fulica* (MN022692) and *Achatina fulica* (Férussac, 1821) (KU365375). The 7274 bp fragment showed a 99.97% of identity with the internal transcribed spacer 2 of a sequence identified as *Subulina octona* (Bruguière, 1789) (MF444887). The longest fragment appeared to be a nearly complete mitochondrial genome. The *cox1* gene was extracted from it, and it showed a 97.86% of identity with *S. octona* (JX988065).

## DISCUSSION

### New records

*Amaga expatria* was described on the basis of two specimens from Bermuda, the holotype, collected in 1988, and a paratype, collected in 1963 (*Jones & Sterrer, 2005*). The species has

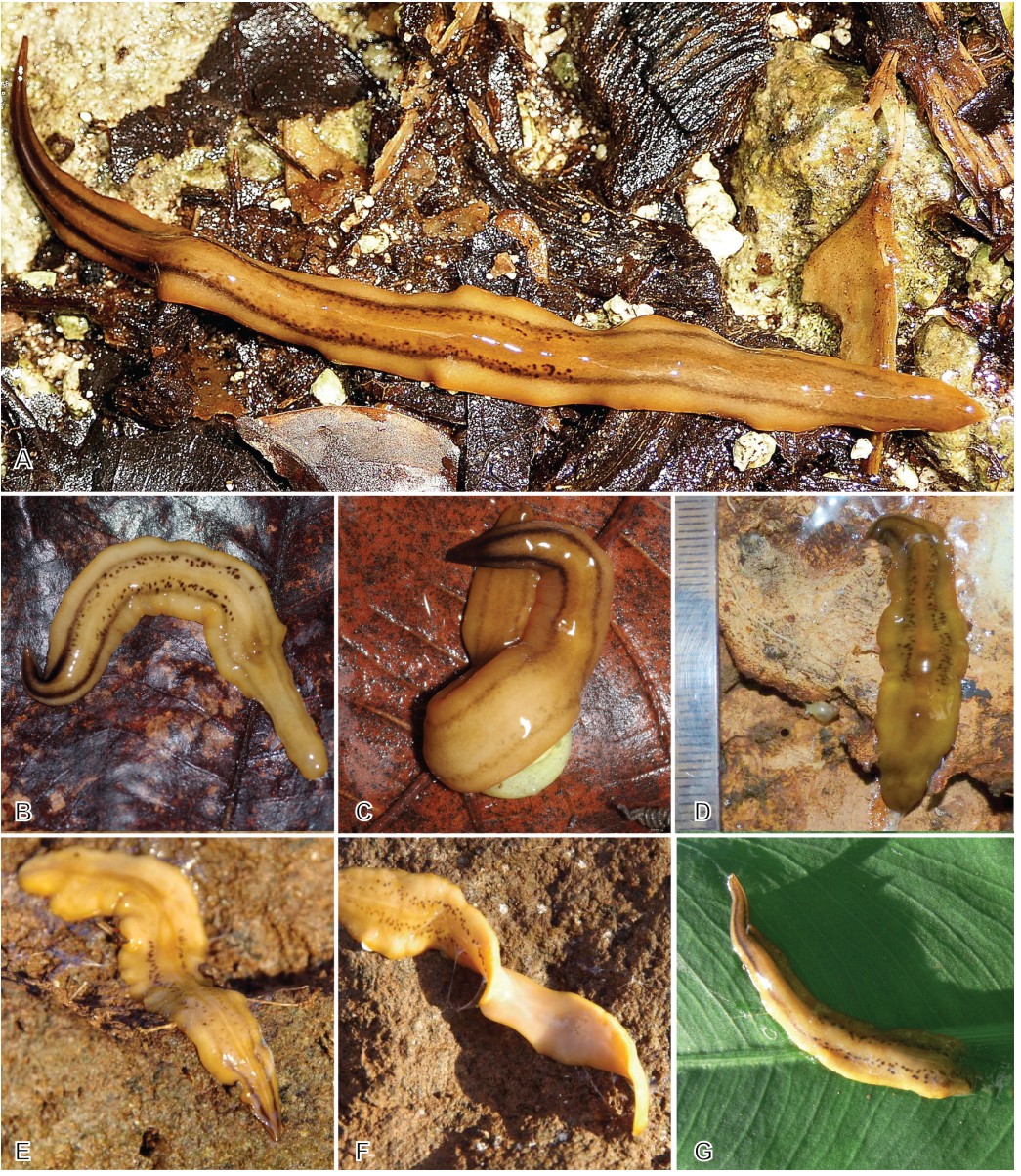

**Figure 5 *Amaga expatria* from Guadeloupe, living specimens in the field.** (A) Photograph by Pierre and Claude Guezennec (anterior tip is left); (B and C) photographs by Laurent Charles, (B) specimen MNHN JL216, (C) MNHN JL217; the prey snail is *Helicina platychila*; (D) photograph by Mathieu Coulis, specimen MNHN JL310; (E–G) photographs by Guy van Laere, (E and F) specimen MNHN JL319, (E) dorsal side, (F) showing ventral side, (G) specimen with damaged posterior part.

not been recorded since, but was mentioned in a book on molluscs of Martinique (*Delannoye et al., 2015*). These were originally identified by one of us (JLJ) and are also included in the present study. Our study has ten times more records than the original description, with 6 records from Guadeloupe and 14 records from Martinique (Table 1). This exemplifies again the power of citizen science for recording land flatworms (*Justine et al., 2018b*, *2019*).

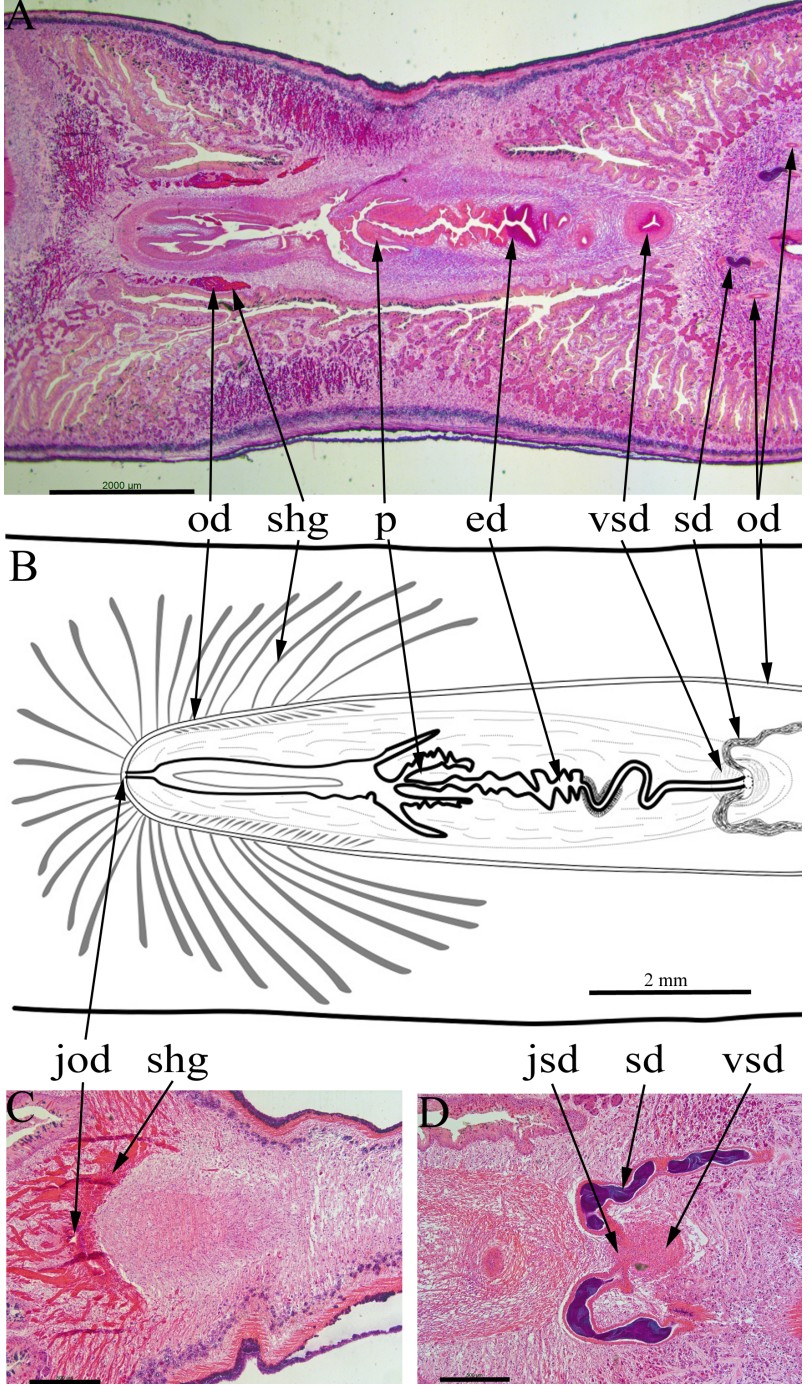

**Figure 6** *Amaga expatria*, **specimen MNHN JL146 from Martinique, anatomy.** Anatomy of the copulatory apparatus, anterior to the right: (A and B) respectively an HLS section and a diagrammatic reconstruction through the copulatory apparatus to the same scale; (C) posterior of the female ducts showing the junction of the ovovitelline ducts; (D) anterior of the male ducts showing the junction of the sperm ducts, both with copious stored sperm (cyanophylic) with the ventral end of the vertical sperm duct. Abbreviations: ed, ejaculatory duct; jod, junction of the ovovitelline ducts; jsd, junction of the sperm ducts and the vertical sperm duct; od, ovovitelline duct; p, penis; sd, sperm duct; shg, shell gland; vsd, vertical sperm duct. Scales: (A and B) 2 mm; (C and D) 500 μm.

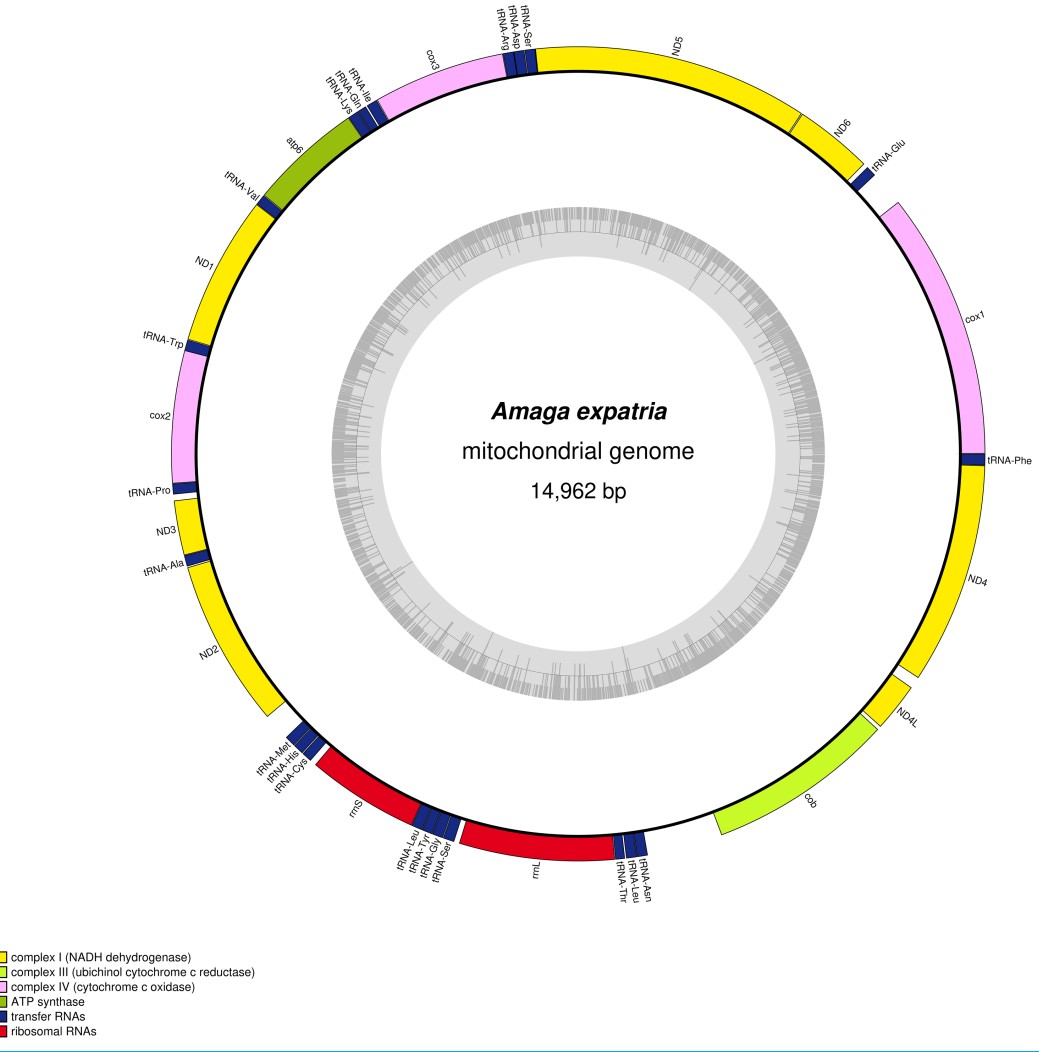

complex I (NADH dehydrogenase)
complex III (ubichinol cytochrome c reductase)
complex IV (cytochrome c oxidase)
ATP synthase
transfer RNAs
ribosomal RNAs

**Figure 7** *Amaga expatria*, **map of the mitochondrial genome.** The mitogenome is 14,962 bp long and contains 12 protein coding genes, two ribosomal RNA genes and 22 transfer RNA genes. For three genes, atp6, cox2 and ND3, it was not possible to determine the start codon.

Maps (Figs. 1 and 2) show that the species is widely spread in both Martinique and Guadeloupe. In Guadeloupe, most records were from Basse Terre and a single record (Sainte-Anne) was from Grande Terre, and in Martinique, most records were from the North of the island, with only one in the South, in Sainte-Luce. This suggests that the species is more abundant in, but not exclusive to, the parts of the islands with higher rainfalls (Basse Terre in Guadeloupe and the North in Martinique).

## Anatomy and morphology

The copulatory apparatus of the Martinique specimen (Fig. 6) is essentially the same as the type specimen of *A. expatria* from Bermuda (NHMUK.2002.10.16.1). The afferent male ducts have a similar structure, with two sperm ducts discharging into a ventro-dorsal duct which in turn opens into the ejaculatory duct and blunt penis via a sinuous duct.

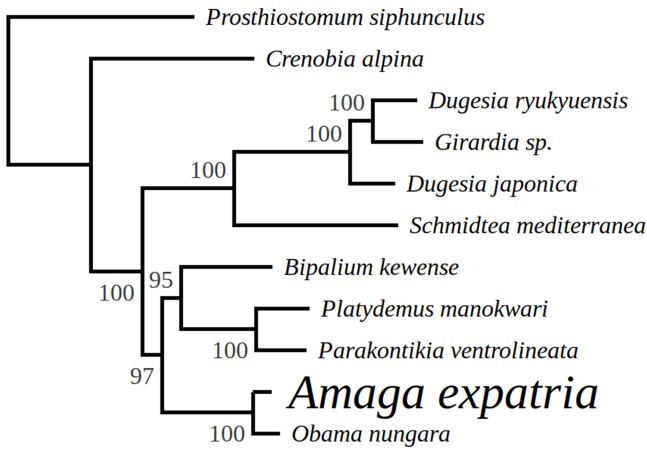

**Figure 8 Maximum likelihood tree of mitogenome proteins.** Mitogenome proteins were obtained from concatenated amino-acid sequences of all mitochondrial protein coding genes of *Amaga expatria* and other Platyhelminthes obtained using the MtArt model of evolution after 100 bootstrap replications. The tree with the best likelihood is shown (−62178.796969).

The duct wall is thickened in the same position about half way between the vertical duct and penis. We are confident of the identification of the Martinique specimen (MNHN JL146) as *A. expatria*. Given this and the similarity of the external characteristics of the specimens from Martinique, Guadeloupe and Bermuda, we are confident that all specimens are *A. expatria*.

## Diet

Analysis of prey DNA is an efficient method to determine the diet of land planarians (*Cuevas-Caballé, Riutort & Álvarez-Presas, 2019*). All BLAST analyses of the contaminant DNA in a specimen of *A. expatria* identified it as belonging to the Gastropoda, and it is likely that the prey was a specimen of *Subulina octona*, or a closely related species. *Subulina octona* is a tropical terrestrial snail, with a cosmopolitan distribution; this mollusc is indeed present in Martinique where it is considered recently introduced (*Anonymous, 2020*).

The original description of *A. expatria* included no direct observation about the diet, but *Jones & Sterrer (2005)*, on the basis of the presence of a plicate pharynx, wrote "it is likely that earthworms are the sole or principal prey of *A. expatria*". The observations by François Meurgey (predation on snails of the genera *Helicina*, *Pleurodonte* and earthworms) and Laurent Charles (predation on *Helicina platychila*) reported here, and the finding of the sequence of a terrestrial mollusc in the gut, indicate a generalist diet, including both molluscs and earthworms. This diet might be one of the reasons of the success of the species to invade various islands in the Caribbean.

## Molecular barcoding

One specimen from Martinique was characterised for morphology, histology, and barcoding and thus represents the first attempt at a molecular characterization of the species. Specimens from Martinique and Guadeloupe provided identical sequences, therefore unequivocally demonstrating that the same species is present on both islands, and, from morphology and anatomy, is *A. expatria*. The absence of genetic divergence between our sequenced specimens suggests that the species was recently introduced into the two islands from a single population. The COI sequence closest to *A. expatria* found by BLAST was *O. nungara*, with a significant difference of 12%. This suggests that the COI sequences can be used for barcoding *A. expatria*, but this should be validated in the future by a comparative study with sequences of other species of *Amaga*, which are currently not available.

## Mitochondrial genome and multigene phylogeny

In the maximum likelihood multigene phylogeny, the clade including *A. expatria* and *O. nungara* has a strong node support of 100, which is congruent with their position in the same Geoplaninae sub-family (Fig. 8). It clearly discriminates them from the Bipaliinae *B. kewense*, the Caenoplaninae *Pa. ventrolineata* and the Rhynchodeminae *Pl. manokwari*.

Among the features conserved between the mitogenomes of *A. expatria* and *O. nungara*, we would like to emphasize the conserved absence of canonical start codons for the three genes *atp6*, *cox2* and *ND3*. Instead, the first amino-acids evidenced from the putative proteins are always a leucine. This leucine is always coded by a TTG codon, except for *A. expatria* where it is replaced by TTA. While no such thing has been evidenced among the recently sequenced mitogenomes of *B. kewense*, *Pl. manokwari* and *Pa. ventrolineata* (*Gastineau & Justine, 2020*; *Gastineau et al., 2019*, *2020*), similar features have also been observed among several Dugesiidae such as *Dugesia japonica* AB618487, *Dugesia ryukyuensis* AB618488 (both in *Sakai & Sakaizumi (2012)*), *Girardia* sp. KP090061 and *Schmidtea mediterranea* NC_022448 and KM821047 (both in *Ross et al., 2016*). The possibility that TTG could act as an alternative start codon was already suggested by *Ross et al. (2016)*. Based on our data, we may suggest that TTA could also be considered. Addressing properly this question might require N-terminal sequencing of these proteins.

We note that *Amaga* and *Obama* belong to the subfamily Geoplaninae, whereas *Platydemus* and *Parakontikia* are members of the Rhynchodeminae and *Bipalium* is a member of the Bipaliinae. This possible variation of the genetic code could thus be limited to a single subfamily within the Geoplanidae, the Geoplaninae.

## CONCLUSION

Our study shows that a relatively large land flatworm species is common in two islands of the Caribbean, and, with 20 new records, adds ten times the previous number of records of the species, which were from a single location, Bermuda, an island located in the

Northeast Atlantic Ocean. *Jones & Sterrer (2005)* hypothesized that the species originated from continental South America and was recently introduced in the Bermuda. Our genetic results show that COI sequences from Martinique and Guadeloupe were identical and thus suggest that the introduction is recent in these islands. It remains that the locality of origin of the species in South America is still unknown. The species preys on molluscs and earthworms and might be a threat to the biodiversity of soil animals, especially molluscs which include endemic and rare species in the Caribbean islands (*Delannoye et al., 2015*). However, no proliferation was recorded and the threat may be minor, but it might also be that *A. expatria* is only in the first stages of invasion and that it will become an invading species in the future; similarly, recent observations have shown that the highly invasive species *Platydemus manokwari* is now invading Guadeloupe (*Justine & Winsor, 2020*). The presence of *A. expatria* in two islands of the Caribbean suggests that it might be present in other islands, and perhaps in continental North America.

## ACKNOWLEDGEMENTS

A complete French translation of the article is available as a Supplemental File / Une traduction française intégrale de l'article est disponible comme fichier supplémentaire: **Le Plathelminthe terrestre Amaga expatria (Geoplanidae) en Guadeloupe et Martinique: nouveaux signalements et caractérisation moléculaire, dont le mitogénome complet.** We are grateful to all non-professionals and professionals who provided records and specimens; they are listed in Table 1. All individuals listed in Table 1 kindly agreed to have their photographs published in this paper. François Meurgey and Laurent Charles kindly communicated some field observations; Laurent Charles provided information about the status of *Subulina octona*; they both agreed to have their information communicated here. Leigh Winsor (James Cook University, Australia) kindly helped in identifying specimens on photograph at early stages of this work.

### Funding

This work was supported by an "Action Thématique du Muséum—ATM" grant from the Muséum National d'Histoire Naturelle, Paris, France. The funders had no role in study design, data collection and analysis, decision to publish, or preparation of the manuscript.

### Grant Disclosures

The following grant information was disclosed by the authors:
Muséum National d'Histoire Naturelle, Paris, France.

### Competing Interests

Jean-Lou Justine is an Academic Editor for PeerJ.

## Author Contributions

- Jean-Lou Justine conceived and designed the experiments, performed the experiments, analyzed the data, prepared figures and/or tables, authored or reviewed drafts of the paper, and approved the final draft.
- Delphine Gey conceived and designed the experiments, performed the experiments, analyzed the data, prepared figures and/or tables, authored or reviewed drafts of the paper, and approved the final draft.
- Jessica Thévenot analyzed the data, prepared figures and/or tables, authored or reviewed drafts of the paper, and approved the final draft.
- Romain Gastineau conceived and designed the experiments, performed the experiments, analyzed the data, prepared figures and/or tables, authored or reviewed drafts of the paper, and approved the final draft.
- Hugh D. Jones conceived and designed the experiments, performed the experiments, analyzed the data, prepared figures and/or tables, authored or reviewed drafts of the paper, and approved the final draft.

## DNA Deposition

The following information was supplied regarding the deposition of DNA sequences:

All new sequences are available at GenBank: MT602619–MT602626, MT527191, MT860713 and MT860719.

## Data Availability

All data are available in the Supplemental Files and from GenBank: MT602619–MT602626. The complete mitogenome is available at GenBank: MT527191. SSU and LSU partial sequences are available at GenBank: MT860713 and MT860719.

All specimens have been deposited in the official collection of the Muséum National d'Histoire Naturelle (MNHN), Paris, France, under registration numbers MNHN JL146, MNHN JL216, MNHN JL217, MNHN JL262, MNHN JL289, MNHN JL305, MNHN JL310 and MNHN JL319.

## Supplemental Information

Supplemental information for this article can be found online at http://dx.doi.org/10.7717/peerj.10098#supplemental-information.

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
