# Peer review of "The land flatworm Amaga expatria (Geoplanidae) in Guadeloupe and Martinique: new reports and molecular characterization including complete mitogenome"

_PeerJ, doi:10.7717/peerj.10098_

## Round 0.1 · original submission · Minor Revisions

Dear Jean-Lou,

The three reviewers are extremely positive about your submitted paper, congratulations. They nonetheless propose a series of minor revisions that I think it is important you follow or else give a rebuttal answer. Be especially attentive to the comments added to a copy of your ms by reviewer 2 (Marta).

I hope to see soon your revised version of the ms.

Best wishes,

Marta

Reviewer 1 ·

Basic reporting

A useful contribution to a growing field cataloguing, describing and developing molecular tools towards understanding land planarians and especially those becoming important invasive pests (for which many have the propensity of becoming). The serendipitous reach of a 'national' citizen science program provided access to various specimens of Amaga expatria for which a mitogenome has been characterized using short molecular barcodes to help verify species identifications.

The writing is clear, professional and high quality. The authors have provided a comprehensive background to the underlying issues and relevant history underpinning the publication. All figures are clear and informative and the article is well executed, with all sections complementing one another in terms of content, interpretation and clarity. I have no concerns with the analysis or interpretation of the data.

Experimental design

The article is novel as the species in question has not been characterised using molecular tools since discovery and description. The delivery of a full mitogenome contributes useful data fro phylogenetics, molecular systematics and potentially diagnostic markers should they be needed. Morphological examination, next generation sequencing, mitogenome mapping for gene boundaries and analysis of NGS data to identify/verify possible food sources are all useful and complementary.

Validity of the findings

All underlying data have been provided; they are robust. Conclusions are well stated, linked to original research investigation and useful for future studies. The TTG/TTA possible start codons are interesting - I do hope GenBank don't have a problem with this (they can be annoying when data don't fit their expectations; good luck)

Additional comments

A few suggested changes in the text:

l.27 we DNA barcoded [to differentiate from 'traditional' barcoding]
l.28 more usually 'next generation sequencing'
l.37 by standard molecular barcoding techniques [as above]
l.173 For 4 specimens, the amplified COI sequences [since the full gene is much bigger than 903 and to differentiate these from the full cox1 gene in the mitogenome]
l.233 BLAST (acronym needs to be capitalised)

Figure 1 and Figure 2 ... indicate annual rainfall [not 'raindrop falls']
Figure 1 and Figure 2 ... where rainfall is high [not 'where raindrops are high']
Figure 7 Amaga expatria, map of the mitochondrial genome [remove 'genomic']
Figure 8 ... I may have missed this; is the phylogeny rooted or unrooted?

·

Basic reporting

Fine.

Experimental design

Fine.

Validity of the findings

Fine.

Additional comments

The article "The land flatworm Amaga expatria (Geoplanidae) in Guadeloupe and Martinique: new reports and molecular characterization including complete mitogenome" is a good contribution to the knowledge of the distribution and characteristics of invasive land planarians, in this case, A. expatria, a species that has been rarely detected so far. The authors present an interesting article on the detection of this species on two islands (Martinique and Guadelupe) giving both morphological and molecular evidence of their identification, and moreover they present a mitochondrial genome of the species, helping to increase the number of mitogenomes available for terrestrial planarians (which is currently scarce).
The figures and the table are informative, necessary, and of good quality.
In summary, I consider that the article can be accepted for publication in PeerJ with some minor changes.
The authors will find my comments and suggestions in the attached manuscript.

Reviewer 3 ·

Basic reporting

See "General comments for the author"

Experimental design

See "General comments for the author"

Validity of the findings

See "General comments for the author"

Additional comments

This paper is an important contribution for the knowledge of land planarians in general. In addition to presenting the complete mitogenome of Amaga expatria, the authors bring important information about diet and some taxonomic aspects (which could be expanded – see comments), complementing the description of the species. In general, the manuscript is clear and concise. Results are solid and informative, same for discussions. Some relatively minor comments and editorial suggestions follow. I do not comment on the language, since I am not a native English speaker. Congratulations to the authors!

Abstract, line 41: Include the author of Obama nungara.

Introduction, line 47: (Geoplanidae) sounds irrelevant, since the information is already in the title. If you want to keep it, I suggest expanding the taxonomic classification to (Platyhelminthes: Geoplanidae), thinking about the public that has no knowledge about land planarians.

Introduction, line 62: “We present here new records of A. expatria…. and provide the first barcoding characterisation”. The work also includes morphological information in its results. It must be mentioned here.

Line 61 and line 69: Gastineau & Justine 2019 and @Plathelminthe4 are not in the references list.

Histology, line 81: Information about number of sections per slide seems irrelevant to be mentioned, since it has no influence in the preparation of slides, nor in the final results. Same on line 84. It could be mentioned how the analyzes of histological sections were performed, changing the section title to “Morphology and histology”, as in the results.

Information obtained from citizen science and other scientists, line 138: Include the author of Helicina platychila.

Morphology and histology: in this section, information on the external morphology of the holotype is transcribed from the original description and after, there is a section “Description of Specimen MNHN JL146.” Since a specimen has had its morphology analyzed, it would be interesting for information about external aspects to be described here as results (and not just body measurements). For example: How is the coloring, eye arrangement, (among other characteristics) in this specimen? If this information is added, it must be included in Material and Methods, as already mentioned.

Description of Specimen MNHN JL146: How is the common muscle coat? By which muscle fibers is it formed? You could add as much information about the species morphology to complement the 2005 description.

Mitogenome, lines 189, 190 and Detection of an alien DNA, lines 202, 203, 205: Include the authors of these species.

Discussion, Anatomy: Did the specimen analyzed reach full maturity? This is an important point to be considered when comparing with the specimens of the original description of the species. Is there any variation in relation to external morphology characteristics? This information could be mentioned in the article!

Figure 1: Map scale is missing.

Figures 3, 4 and 5: Whenever it is possible to calculate a scale, please add it in the photos with a bar indicating the scale value and preferably standardized. Specimens that cannot have a calculated scale (due to not really knowing the size of the specimen, not even approximately) must be informed in the legend: “Scale bar for (XXX) not available”. Likewise, it would be better to standardize the location of the photo's indication (A, B, C….), always in the same position in all photos. Another point to be commented on the photos: the anterior tip of specimens must always be positioned on the same side, informing in the caption “Anterior tip to the left; or on the top…”. This facilitates the reader's understanding, especially if the reader is not familiar with land planarians. This standardization of information of the anterior tip may also be performed with an arrow, for example.

Figure 5C: Please, point in the photo the prey snail Helicina platychila.

Figure 5F: The photo shows the dorsal side and ventral side. It would be important to indicate the face to be observed.

Figure 5G: The posterior end was accidentally lost? It is not clear in the photo.

Figure 6: The indications (A, B, C and not a, b, c) must be standardized with the other photos in the article.

Figure 6B: It must present a scale.

Figures 6C, D: The scale value is not visible.

Figure 7: It could be improved so that the indications are visible.

Figure 8: "sp." is in italics?

---

## Round 0.2 · Minor Revisions

Dear Jean-Lou,

I sent for revision your new versions and rebuttal comments to two of the original reviewers. One answers that they are satisfied with your changes but the other, on maternity leave, has asked me to take her place and do the re-revision. I have checked her original revision and your answers and I have a few minor revisions that I think need to be done.

In the first place there is something that I see none of the reviewers has asked you and I'd like to know (maybe I have missed the point), but in the ms in line 105 it is stated: "Contigs corresponding to the mitogenome and the nuclear ribosomal RNA genes were retrieved"
What was the use given to the nuclear ribosomal RNA? I do not see it mentioned any more in the ms (as I said, I must have missed it). Were the sequences, nonetheless, uploaded to GenBank and given an accession number? It will be interesting that they are public, if they exist, for the sake of future works. So, please, explain what did you retrieve those sequences for, and specially upload them to GenBank and give the accession numbers somewhere in the present paper.

Line 188: The best result are listed thereafter

Make plural “results” (or use "is")

In the description of the trees (Line 244, and I think somewhere else) you use the word "associate" to describe some relationships. To me “associate” seems a not very professional word to use to describe a phylogenetic tree, in this case it will be better to say “… phylogeny shows A. expatria to be the sister group of Obama nungara…” (or “... phylogeny shows A. expatria to constitute a monophyletic group with…”). Please, change.

One comment:
Line 251. Why this 12% of genetic distance is related to the validity of the gene to be used as barcoding? Actually, it has been already tested in land planarians that this molecular marker is useful for molecular identification, but it is necessary to include the sequences in a phylogenetic framework. The 12% only suggest that O. nungara and A. expatria are well differentiated species, but the presence of more genera is necessary to check if this is really A. expatria or not (like a proper barcode). Unfortunatelly, this is the first record of this genus (and species) in molecular databases, and it is impossible to test 100% identity. However, a more closely relationship to other genera can be tested with molecular phylogenies or even genetic distances.

Your rebuttal:
The purpose of this paper was to demonstrate that the same species was found in two islands. COI barcoding was used as a way to delimitate species, certainly not to establish a phylogeny at the genus level. We tested our new COI sequences in GenBank BLAST and found that the most similar sequence was one from Obama nungara, and therefore we used this sequence; it happens that this sequence has a 12% difference with A. expatria, meaning that other available sequences, from other species, have a higher divergence. Our sequences are now available for studies on phylogeny by other authors – again, this was not the purpose of this study. Also, 28S and 18S sequences would probably be better for phylogeny. No change made.

My new comment:
In this case I think that what the reviewer was trying to say is that the sentence “The difference of 12% with O. nungara suggests that the COI sequences can valuably be used for barcoding A. expatria” does not make sense as it is. It is true that it has already been shown that COI is a good DNAbarcoding tool for terrestrial planarians. In the present case it is clear that the absolute lack of differences in the sequences of this gene among individuals from the different islands demonstrates all belong to the same species. But the sentence stating that finding a 12% difference between O. nungara and A. expatria demonstrates that COI is a valuable barcoding tool for A. expatria is not true. It might demonstrate that it can be useful to differentiate the two genera, but to state that COI is a good dnabarcoding tool for A. expatria it will be necessary to have more representatives of the genus Amaga and show that they can be differentiated (delimited) as different species by the analysis of their COI sequences.
In conclusion, remove the sentence from the discussion.

And just a comment on some of the answers to the third reviewer. I agree in that we regularly do not give the authories of the species that we use or cite in our papers. And when we do, we only cite them in the text but give not the reference in the reference list (some may be very old, yes). However, having into account the recent problem with zootaxa and other mostly taxonomic journals and the impact factor, I think this should be reconsidered. In my view, not only the authories of the species must be given, but also it will be necessary that the papers included in the reference list. We include the authors of the primers we use, giving credit and many cites to some papers (one of my most cited papers if for that, for giving a list of primers), while we do not give credit to the work of the people who delimited and gave name to that entity with which we are now working, if we think it seriously it does not make any sense. Anyway, this was simply a reflection.

And that's all. Send the new version with these few changes and I'll readily accept the paper. Congratulations for a great work.

Best

Marta

Reviewer 3 ·

Basic reporting

'no comment'

Experimental design

'no comment'

Validity of the findings

'no comment'

Additional comments

'no comment'

---

## Round 0.3 · accepted · Accept

Dear Jean-Lou,
Sorry for the delay, I had a deadline to present the justification for a project tomorrow, and so forgot about many other things. Now done, congratulations on the new paper.

Marta